# Decoupling RNN Training and Testing Observation Intervals for Spectrum Sensing Applications

**DOI:** 10.3390/s22134706

**Published:** 2022-06-22

**Authors:** Megan O. Moore, R. Michael Buehrer, William Chris Headley

**Affiliations:** 1Hume Center for National Security and Technology, Virginia Tech, Blacksburg, VA 24061, USA; cheadley@vt.edu; 2Department of Electrical & Computer Engineering, Virginia Tech, Blacksburg, VA 24061, USA; rbuehrer@vt.edu

**Keywords:** modulation classification, radio frequency machine learning, recurrent neural networks, spectrum sensing

## Abstract

Recurrent neural networks have been shown to outperform other architectures when processing temporally correlated data, such as from wireless communication signals. However, compared to other architectures, such as convolutional neural networks, recurrent neural networks can suffer from drastically longer training and evaluation times due to their inherent sample-by-sample data processing, while traditional usage of both of these architectures assumes a fixed observation interval during both training and testing, the sample-by-sample processing capabilities of recurrent neural networks opens the door for alternative approaches. Rather than assuming that the testing and observation intervals are equivalent, the observation intervals can be “decoupled” or set independently. This can potentially reduce training times and will allow for trained networks to be adapted to different applications without retraining. This work illustrates the benefits and considerations needed when “decoupling” these observation intervals for spectrum sensing applications, using modulation classification as the example use case. The sample-by-sample processing of RNNs also allows for the relaxation of the typical requirement of a fixed time duration of the signals of interest. Allowing for variable observation intervals is important in real-time applications like cognitive radio where decisions need to be made as quickly and accurately as possible as well as in applications like electronic warfare in which the sequence length of the signal of interest may be unknown. This work examines a real-time post-processing method called “just enough” decision making that allows for variable observation intervals. In particular, this work shows that, intuitively, this method can be leveraged to process less data (i.e., shorter observation intervals) for simpler inputs (less complicated signal types or channel conditions). Less intuitively, this works shows that the “decoupling” is dependent on appropriate training to avoid bias and ensure generalization.

## 1. Introduction

Typically in radio frequency machine learning (RFML) (RFML is a field in which machine learning is applied to solve problems in radar, signal intelligence, electronic warfare, and communications systems) applications, time is the most constrained resource during both training and inference. During the training process, the needs of both dataset collection/generation and architecture optimization increase as the amount of sequential observation data needed for the algorithm to reliably generalize the problem increases. This increase occurs with respect to both computational time and man-power and is typically a function of the complexity of the application. The amount of data needed for a reliable output is dependent on the input signal parameters and RF channel propagation conditions and is not constant. For example, the authors’ prior work demonstrated that simpler input formats with good channel propagation conditions (higher signal-to-noise ratios (SNR), lower frequency offsets, simpler modulation schemes, etc.) required lower sequential observation data needs to make a reliable decision [1]. Once fielded, many applications are extremely time sensitive and can greatly benefit from processing as little data as necessary to make a reliable decision. Therefore, for many RFML applications, fixed observation intervals are sub-optimal and an unrealistic assumption.

In this paper, the inherent sample-by-sample nature of recurrent neural networks (RNN) will be utilized to test the impact of setting training and testing sequence lengths independently. This “decoupling” relaxes the assumption of equivalent input sequence lengths that has previously been used and allows for the potential of greater flexibility between training and inference. Consider a network that is trained on short, fixed observation intervals due to limited time for dataset collection. When the network is deployed, it may need to process signals that are longer than it ever saw in training. Similarly, a network trained for an application with one sequence length should be able to be applied to an equivalent application with a different sequence length—without retraining the network. Ref. [2] addressed this problem for convolutional neural networks (CNN) by processing multiple, fixed observation intervals and fusing the results. However, with RNNs, a variable sequence length can be easily processed with no changes necessary to the architecture. Ref. [3] tested on sequence lengths longer than those used in training, however, they did not examine how disparate training and testing sequence lengths impacted network bias, generalization to data outside the training range, and performance for different channel conditions. By examining the trends of training and testing sequence lengths, the most advantageous sequence length for a problem set can be selected—potentially reducing training time and network complexity and allowing for increased flexibility in deployment.

In many cases, the sequence length in inference may be variable or completely unknown. Rather than setting a fixed testing sequence length it would be preferable to allow for variable sequence lengths to be processed. Such an approach is commonly used in machine text translation where the network will process inputs until it displays an <EOS> (end of sentence) key [4]. However, in spectrum sensing applications the choice of appropriate evaluation lengths is not nearly as obvious—especially with limited or no prior knowledge of the signals of interest, while pre-processing steps could be added to identify and separate the signal prior to feeding it into the network, this would not be ideal for real-time deployments. In time-sensitive applications like electronic warfare, radar, and dynamic-spectrum access (DSA), decisions need to be made as quickly and accurately as possible. To allow for faster processing, many researchers have focused on reducing network complexity [5,6,7]. However, these approaches result in processing every signal for the same amount of time—regardless of signal complexity. Consider the case discussed in [8] where signals have different oversampling rates. The signals with higher oversampling rates needed to be processed for longer in order to see enough complete symbols to correctly identify the class. However, without knowing the oversampling rate of the signal ahead of time, it is unclear when to stop processing inputs. In our initial paper [1], we introduced the “just enough” decision making (JED) method—an approach that dynamically chooses how much sequential input data to process based on signal complexity. However, the approach was only tested for a single training sequence length and did not allow for signals to be processed for longer sequence lengths. The JED method will be evaluated on additional parameters to determine its true utility.

The contributions of this paper can be summarized as the following:Verified that the performance of a neural network based modulation classifier is bounded only by the number of symbols seen in inference.Examined the impact of “decoupled” training and testing sequence lengths on network bias and generalization.Demonstrated that the JED method can improve accuracy and reduce the average number of samples processed.Showed that the JED method chooses how much sequential input data to process based on signal complexity.

While RNNs have seen use in applications such as RF fingerprinting [9], spectrum prediction [10,11], and signal classification [12,13,14], the scope of this paper’s analysis is an RNN-based Automatic Modulation Classifier (AMC). Although much work has been done on AMCs, it is still an important topic in spectrum sensing with recent applications in MIMO systems [15,16], while modulation classification will be used for the initial proof of concept, the results are expected to generalize to other areas, and a similar analysis for signal detection can be found in Appendix 2 of [17].

In Section 3, the data sets and networks used in this paper are introduced. In Section 4, the maximum likelihood (ML) classifier is introduced and compared to networks trained on different observation intervals. In Section 5, the potential benefits and detriments of training on longer observation intervals are identified. In Section 6, the JED method is examined for different nuisance parameters and evaluated for its ability to differentiate between simple and complex signals. Section 7 present a brief summary and discussion of the results. Finally, this work is concluded and future work is suggested in Section 8.

## 2. Background

Traditional approaches for spectrum sensing applications have relied on expert knowledge and feature extraction or simplistic models. As such, they often require a human operator and have strong assumptions concerning a priori knowledge [18]. For example, common signal detectors include matched filter detectors that require perfect knowledge of the signal, energy detectors that are sensitive to noise, and cyclostationary feature detectors that are computationally expensive [19]. In contrast to traditional approaches, the data driven approach of deep neural networks (DNN) allows feature extraction to take place internally and can reduce the number of required assumptions. Current DNN approaches to RFML include both supervised and unsupervised approaches, however, supervised networks are more commonly seen since they are trained to perform a specific task rather than just search for structure in the data. Prior work in RFML has commonly leveraged CNNs [20,21], but RNNs and hybrid CNN-RNN networks have become more common over the last decade.

RNNs are a unique class of neural network defined by their ability to process inputs sequentially. They rely on the concept of a cell that is continually updated with new inputs and a hidden state allowing. As a result, they can be considered an Infinite Impulse Response (IIR) system [22]. The cell contains the actual network weights which are shared across all time inputs; this allows cells to be copied as many times as necessary to process an entire input sequence.

Vanilla RNNs suffer from instability due to what is commonly referred to as the “vanishing and exploding gradient problem” [22] which makes it difficult to learn long-term dependencies. To combat this problem, a new network known as the Long-Short-Term Memory (LSTM) network was developed in 1997 [23]. The LSTM network consists of three gates in each cell: the forget gate, input gate, and output gate. The forget gate considers what information should be thrown away from the hidden state, the input gate considers which values will be updated with the new input, and the output gate determines the next hidden state. Each gate consists of a sigmoid function to keep gradients within the range of 0 to 1. Another common type of RNN, the Gated Recurrent Unit (GRU), is an LSTM variant that combines the input and forget gate resulting in fewer parameters and faster processing with minimal performance loss [24].

Since LSTMs and GRUs can learn the long-term dependencies seen in RF signals, they are the most commonly used types of RNNs in RFML. They have shown incredible promise for their usage in applications such as RF fingerprinting [9], spectrum prediction with Cognitive Radios [10,11], and modulation classification [12,13,14], among others.

Due to their serial processing nature, RNNs have been shown to outperform other commonly seen networks like CNNs in some spectrum sensing applications. In [25], a 2-layer LSTM model given amplitude phase inputs outperformed the CNN model described in [26] by over 10% in both the low and high SNR range. In [12], a 1-layer LSTM model outperformed the same CNN model by 6% for SNRs greater than 4 dB when using raw I/Q inputs.

Although RNNs can improve performance in some cases, they tend to be very time consuming to train and evaluate—particularly for longer sequences. In one comparison, an LSTM network took 75 times as long to train and twice as long to evaluate each sample when compared to a CNN [27]. As such, hybrid networks that consist of both RNN and CNN layers [28,29], approaches that mimic RNNs like dilated CNNs [30,31], and sequential CNNs [27] have been introduced. Many of these networks have performance that is equivalent or superior to RNNs. For example, in [32] a CNN, LSTM, and a convolutional LSTM deep neural networks (CLDNN) were all trained to perform AMC under Rayleigh fading. The LSTM and CLDNN performed similarly under Rayleigh fading, but CLDNN outperformed the LSTM under AWGN conditions.

It is also important to remember that accuracy is not the only consideration when choosing a network architecture—the utility of the model must also be considered. For example, in [7] a CNN and LSTM had similar accuracy, but the CNN was preferred due to having fewer trainable parameters. However, while CNNs and hybrid networks may be preferred in some scenarios, they typically assume a fixed block input size which is not desirable for many real-time scenarios. Ref. [2] addresses the idea of processing a longer sequence than trained for with a CNN by processing the signal in sections and fusing the results, however, the method is limited in its flexibility. Unlike CNNs which are fed a block of inputs and are typically restricted to a fixed input size, RNNs can be extended indefinitely. This allows for networks to be trained on varying sequence lengths and to be evaluated for any arbitrary number of input samples.

In many spectrum sensing cases, a variable sequence length is advantageous. For example, in DSA interfering signals need to be identified as quickly as possible so that the interference can be mitigated. To allow for the faster processing required in some applications, performance may be sacrificed. Ref. [3] improved the performance of their AMC by testing on a longer sequence length than used in training, however, they felt that the increased computational time was too large a detriment. By using a fixed sequence length, they were forced to process every signal for the longer sequence length instead of only those that would most benefit from further processing. Simpler input formats with good channel propagation conditions (higher signal-to-noise ratios, lower frequency offsets, simpler modulation schemes, etc.) may require lower sequential data needs to make a reliable decision.

The true utility of RNNs is the ability to easily process a variable number of samples. In [8], an RNN based AMC network was trained on three separate sequence lengths determined by the oversampling rate. Signals that had a higher oversampling rate needed to be processed for additional samples in order to see the same number of symbols. However, it was unclear how this would be approached if the oversampling rate of an input signal was unknown. Ref. [5] recognized the potential for real-time classification with LSTMs by training the classifier only on the final hidden state of an LSTM autoencoder. The new training process significantly reduced training time. It also allowed for faster real-time classification as only the LSTM operations were repeated at each time step. However, the paper did not address how to determine a stopping condition in inference.

In general, longer sequence lengths—in either training or testing—will result in improved performance [6,33]. However, a thorough analysis of the impact of training and testing sequences—individually and combined—has not been performed. In particular, the impact of sequence length on network bias and generalization to data outside the original training range need to be considered.

In [1] we, proposed a real-time post-processing decision making method. It was an initial attempt to dynamically alter the testing sequence length based on the output of the network’s softmax value over time. The approach should be able to handle variable or unknown signal lengths by continually processing inputs until a decision criteria is reached. However, it was not tested on a larger number of samples than seen in training or for networks trained on different sequence lengths. Further analysis will be performed to test the efficacy of the approach.

## 3. System Model

For the analysis that follows, three spectrum scenarios are considered which dictate the training assumptions of the RNNs. The first scenario assumes perfect synchronization and matched filtering between the transmitter and receiver, allowing for the use of received symbols as input to the RNN. This scenario, termed **symbol** in this work, will act as our baseline performance given optimal sensing conditions. The second scenario, termed **sample** throughout this work, assumes frequency synchronization and Nyquist sampling, but not matched filtering. Therefore, in this scenario, the input to the RNN is the received IQ samples, not symbols. Finally, the third scenario, termed **nuisance**, assumes no synchronization or matched filtering and considers additional nuisance effects such as frequency offsets and greater than 2 times oversampling. The final scenario is more indicative of real-world sensing conditions in which the sensor is non-cooperative with the transmitter.

### 3.1. Data Generation

For each of the three sensing scenarios, synthetic signals were generated for five modulation schemes under test, namely BPSK, QPSK, 8PSK, 16QAM, and 64QAM. The takeaways provided in this work are not expected to be dependent on this specific class of signals. These modulations were chosen for their simplicity of synthetic generation, ease of defining a theoretical bound, as well as their variability in difficulty. For example, consider that BPSK should be easily identified after a few samples while differentiating between 16QAM and 64QAM is difficult for shorter sequence lengths. Table 1 shows the data generation parameters used for each considered spectrum scenario. For all scenarios, the signals were pulse shaped with a root-raised-cosine filter with a roll-off factor of 0.35. The propagation channel for each scenario is assumed to be AWGN with a random SNR between 0 dB and 10 dB. Future work will include training on additional modulations and more complex channel models.

### 3.2. RNN Model Architecture and Training Process

The general network architecture in this analysis is shown in Table 2. It consists of a variable number of LSTM layers, a dropout layer to prevent regularization, and then two fully-connected layers with different activation functions. The architecture is similar to the ones presented in [7,34] which consist of multiple LSTM layers followed by two dense layers. Any sequence-to-sequence classification network that operates on each time-step independently could reasonably be used for this analysis. For example, replacing the first dense layer with a 1-d convolutional layer would still allow for variable length inputs to be processed. Figure 1 shows a time-unrolled version of the architecture—including the passing of the hidden state, while the network is technically sequence-to-sequence, only the output of the final time step $Y_{N}$ is used when calculating accuracy.

Each network was trained for a maximum of 30 epochs with a training/validation split of 80/20. The Adam optimizer was used with a fixed learning rate of 0.001. Different fixed sequence lengths were examined to determine how they impacted the generalization of the network. However, based on the findings of this work, future work should include training on variable sequence lengths.

For each scenario and sequence length, five random architectures were trained. Multiple architectures were used in order to ensure the results were not dependent on or—specific to—a single architecture. As this was not meant to be an exhaustive hyper-parameter search, not all parameters were varied. The bounds of the parameters that were varied were chosen based on architectures seen in the literature. The most commonly seen hidden state size was 256 and the most common number of LSTM layers was 2—values above and below these were chosen [3,25]. The number of LSTM layers is set as *l*—a randomly chosen integer between 1 and 4. The hidden size of the LSTM layers was set as *h* and the size of the first dense layer was set as *d* where *h* and *d* are randomly chosen integers between 15 and 512. The input size of each LSTM cell is 2 to accommodate both the in-phase and quadrature data, and the hidden size represents the size of the hidden state in each cell. When processing a sequence, each sample is processed by the LSTM cell and the hidden state is updated allowing for arbitrary sequence lengths.

The total number of trainable weights in the network can be calculated as:(1)$$\begin{array}{c} W=4h(2hl-h+l+2)+hd+5d. \end{array}$$

As network weights are used for each time-step, the total number of matrix multiplications required will be W∗n where n is the sequence length of the input. Using big-O notation, the time-complexity can be written as O(n). The time-complexity of the network grows linearly with the input sequence length.

The same random architectures were not used for each sequence length since the optimal architecture may vary with the length of the input sequence. The performance of each network was examined, but for brevity, only the results for the best performing network of each sequence length will be shown. Plots for the other networks can be found in the Appendix of [17]. The best network for each sequence length was determined by finding the network that had maximum probability of correct classification (PCC) when tested on the same sequence length it was trained on. The **symbol** models were trained on sequence lengths of 64, 128, 256, and 512 symbols and compared to the **sample** models on trained sequence lengths of 128, 256, 512, and 1024 samples (ensuring an equivalent number of observed symbols between these first two scenarios). The **nuisance** models were also trained on sequence lengths of 128, 256, 512, and 1024 samples. Note that the samples and symbols are normalized to unit average energy, so that they can be accurately compared regardless of sequence length and oversampling ratio.

## 4. Symbol Input Scenario

Prior work formulated the optimal ML approach for classification of digital amplitude phase signals. Assuming perfect synchronization (The assumption of perfect synchronization is not realistic for extremely low SNRs and short observation lengths leaving the findings of this work mostly as an upper bound on performance and not a realistically achievable goal in practice), matched filtering, and an AWGN channel, the observed symbols are the sufficient statistic of the received IQ data [35]. Under these conditions, for a given SNR and number of observed symbols, the PCC of the ML classifier can be determined. Since the ML classifier has the lowest error rate of all classifiers based on complex domain data, its PCC acts as the theoretical upper bound of the trained AMCs. While the ML classifier is optimal under the specified conditions, it has high computational complexity leading other approaches to be preferred [36].

In order to classify *c* different constellations—each consisting of a set of $M_{j}$ symbols *S*—a test on the following hypothesis is used:(2)$$H_{j}:\{S_{j,0},S_{j,1}\ldots S_{j,M_{j}}\}\;j=1,2,\ldots ,c.$$

The received IQ stream – $X_{N}$ – consisting of *N* symbols and SNR=A can be defined as follows:(3)$$\begin{array}{c} X_{N}=\left\{x_{k}=\left(r_{I,k},r_{Q,k}\right),\;k=1,2,\ldots ,N\right\}. \end{array}$$

In order to classify $X_{N}$, the ML classifier predicts $H_{j}^{*}$ so as to maximize the log-likelihood, $L(H_{j}|X_{n})$. This can be expressed as the following:(4)$$\begin{array}{cc} H_{j}^{*} & =\arg\;\max_{H_{j}}\;\sum_{k=i}^{N}\ln\sum_{i=1}^{M_{j}}\exp\left(-\frac{1}{2}\left|\right|x_{k}-AS_{j,i}{\left|\right|}^{2}\right). \end{array}$$

To find the PCC for the ML classifier, a Monte-Carlo simulation consisting of 1000 random signals for each modulation and SNR.

Figure 2 shows a comparison between the best trained RNN model for the **samples** scenario, the best trained RNN model for the **symbols** scenario, and the ML bound—labeled “Theory”—for different numbers of received symbols. In the case of the trained models, the plotted model was chosen by comparing the overall average PCC when tested on the same number of symbols that the network was trained for. These plots show the average PCC for each SNR. For clarity, only the results for 64 and 512 symbols are shown, but 128 and 256 were also tested and followed similar trends.

The network trained and tested on 64 symbols achieved a little below the theoretical maximum for 64 observed symbols. However, when the network observed 512 symbols, performance improved beyond the theoretical maximum for 64 symbols. Clearly network performance is not bounded by the maximum performance of the training sequence length. However, it is still bound by the maximum performance of the testing sequence length. Additionally, the network trained and tested on 512 symbols was much closer to the maximum than the network trained on 64 symbols and tested on 512 symbols—particularly for lower SNRs. This suggests that training on a sequence length much shorter than that used during testing could result in decreased performance.

Surprisingly, the **samples** networks had higher overall performance than the **symbols** networks. Considering that observed symbols are the sufficient statistic for this scenario, it seems that for the same number of observed symbols the two should have had equivalent accuracy. The difference could be due to different random architectures. To test this, 20 identical random architectures were trained for both **symbols** and **samples** on 64 symbols. When comparing the same network architectures, with the only difference being the sequence length (64 when trained on symbols and 128 when trained on samples), the **samples** networks outperformed the **symbols** networks every single time. One possible explanations for this behaviour is that RNNs have better performance on longer sequence lengths. Another possible explanation is that RNNs are better at finding correlation in samples rather than symbols. This behaviour will be discussed more when examining the impact of different oversampling rates in Section 5.

Overall while the number of symbols used in training is relevant, the number of symbols observed in evaluation has more influence on network performance. For a sufficiently trained classifier, network performance can continue to improve as more observed symbols are processed over what was used during training. What determines if a network is sufficiently trained depends on the problem set and will be discussed more in Section 5.

## 5. Training Sequence Length

When training RNNs, the choice of the training sequence length is often overlooked. Shorter sequence lengths are typically preferred due to how time-consuming training longer sequence lengths can be. As shown in Section 4, networks trained on a short sequence length can improve their performance by processing more samples in evaluation. However, the resulting networks may not be as successful as networks trained on longer sequence lengths. With this in mind, this section aims to help determine how to choose the training sequence length for a problem set.

To examine the impact of sequence lengths in training, the **nuisance** networks were trained and evaluated on various sequence lengths. Figure 3 shows the best model for each trained sequence averaged over all classes. Increasing the observed sequence length increased the PCC with diminishing returns. When trained on 128 samples, the networks reached a point where additional input samples no longer increased performance. However, for the networks trained on longer sequence lengths, the slope of the line suggests that observing additional samples would continue to increase performance. Based on the results found in Section 4, as long as more input samples are given, the classification accuracy should continue to increase if the network has been sufficiently trained. Sufficiently trained does not refer to a specific accuracy, rather it refers to the network’s ability to generalize.

Consider the number of examples used in training a network. The more complex the problem, the more examples the network needs to see to adequately handle not only inputs similar to those seen in training, but also those outside the training range. Sequence length is similar. If the network was trained on a single sample or symbol, it would be unable to generalize to longer sequence lengths and would be an extremely poor network. Based on the lack of performance improvement for the networks trained on a smaller sequence length, they do not seem to be sufficiently trained. It is worth noting that all networks used the same number of examples no matter their sequence lengths. Future work will examine the relationship between training with more data and using a shorter sequence length so as to keep the number of overall samples seen in training constant.

Although the networks trained on 128 samples did not continue to increase their performance when tested on longer sequence lengths, they outperformed the networks trained on 1024 and 512 samples when tested on shorter sequence lengths. However, it is worth noting that while the network trained on 128 far outperformed the 1024 network when evaluated on 128 samples for 10 dB, there was almost no difference at 0 dB. This suggests that for more complex problems, there is less detriment to training on a longer sequence length and evaluating on a shorter one. To further examine this, the effect of modulation, SNR, frequency offset, and oversampling rate at different sequence lengths is examined. When each parameter is examined, the other parameters are varied uniformly across the training range. The parameters under consideration were chosen based on the study in [37]. The study has additional simulations showing the impact of particular parameters when others are held constant.

In the following, the performance plots shown depict the performance of the best randomly trained models. Parameters outside the training range were tested when possible, while the **nuisance** networks were trained on sequence lengths for 128, 256, 512, and 1024 samples, for brevity, only the plots for 128 and 1024 trained samples will be shown.

### 5.1. Modulation

Due to observing fewer unique symbols, training on a smaller sequence length could result in a higher likelihood of model bias and lack of proper generalization. To investigate the potential for network bias, the $F_{1}$ score was examined for all trained **nuisance** networks. The $F_{1}$ score is commonly used in classification problems as it averages the precision and recall for a network. Unlike PCC which only considers true positives, it takes into account false positives and false negatives as well. The $F_{1}$ score can be considered a distillation of the confusion matrices commonly seen in modulation classification problems. Rather than examining which class a signal was misclassified as, the $F_{1}$ score considers only that it was misclassified. Like PCC, the $F_{1}$ score lies in the range of 0 to 1 with 1 being optimal. We can define tp (true positive) as the number of times that a signal of a given class was correctly identified, fp (false positive) as the number of times that a signal was incorrectly identified as a given class, and fn (false negative) as the number of times that a signal of a given class was incorrectly identified as another class. The equation used for calculating the $F_{1}$ score is defined as the following:(5)$$\begin{array}{cc} F_{1} & =\frac{2*tp}{2tp+fp+fn}. \end{array}$$

The $F_{1}$ score for models trained on sequence lengths of 128, 256, 512, and 1024 and then tested on a sequence length of 128 and 1024 samples were averaged over all models evaluated. The scores were averaged to prevent the bias of a single network dominating the results. As no clear trend was found between training on fewer samples and a likelihood of network bias, the results for this analysis are not shown.

However, although training on smaller sequence lengths does not appear to result in a higher likelihood of bias, that does not mean that network bias is altogether unaffected by sequence length. Table 3 shows the $F_{1}$ score of a biased network trained on 128 samples and a biased network trained on 1024 samples. The first two rows show the $F_{1}$ score of a network trained on 128 samples while the last two rows show the $F_{1}$ scores of a network trained on 1024 samples. When tested on 128 samples, the bias is not very noticeable. The network trained on 128 samples slightly prefers 64QAM over 16QAM while the network trained on 1024 samples slightly prefers QPSK to 8PSK. However, when tested on 1024 samples, the $F_{1}$ score shows a significant bias. The $F_{1}$ score for the network trained on 128 samples actually drops for 16QAM and increases substantially for 64QAM. An even more dramatic result occurs for the network trained on 1024 samples, with the $F_{1}$ score for 8PSK decreasing by half while QPSK increases. When testing a biased network on a longer sequence length, regardless of the sequence length used in training, the bias actually becomes more apparent.

Figure 4 shows the impact of different training and testing sequence lengths for each modulation. For the network trained on 128 samples (This is not the biased network used to generate Table 3. The $F_{1}$ score for this network increased for each signal type when tested on a sequence length of 1024), performance does not increase for each signal type when the testing sequence length increases, while the accuracy of 8PSK and 16QAM increase substantially, BPSK, QPSK, and 64QAM basically do not improve at all. This is in contrast to the network trained on 1024 samples in which the performance of each class increased when more samples were tested. Comparisons of the other trained models show similar results. This suggests that the networks trained on a smaller sequence length did not adequately learn to identify each modulation and were unable to generalize to longer sequence lengths.

### 5.2. SNR

To examine the impact of training and testing sequence lengths for different SNRs, values inside and outside the training range were tested. The gray dashed lines in Figure 5 show the training range. The network trained on 128 samples performed very poorly when given SNRs lower than 0 dB. At −5 dB, the network had performance equivalent to random guessing. At SNRs over 10 dB, the network showed very little improvement. In contrast, the network trained on 1024 samples showed improved performance for SNRs outside the training range, particularly those below 0 dB. Overall, this suggests that the networks trained on longer sequence length are better able to generalize to unseen data, even for sub-zero SNRs.

### 5.3. Frequency Offset

Frequency offset can significantly degrade classification accuracy, in [37], a frequency offset of just 2.5% of the sampling frequency reduced the testing performance by over 10%. To examine the impact of training and testing sequence lengths for different normalized frequency offsets, values inside and outside the training range were tested. Again, the gray dashed lines show the training range. The plots in Figure 6 are symmetrical with a plateau around zero. As the frequency offset approaches zero, the gap between the performance of different training sequences seems to increase. The values outside the training range that were tested, ±5%, performed poorly—around 30% when trained on 128 and tested on 2048 samples. However, when trained on 1024 samples, performance improved to almost 40% when tested on 2048 samples. Again, this suggests that training on a longer sequence length allows the network to better generalize to unseen data.

### 5.4. Oversampling Rate

Like frequency offsets, varied oversampling rates—especially ones not seen in training —can degrade performance. Ref. [37] demonstrated that testing on oversampling rates not seen in training quickly degrades performance while accuracy across the training range tends to be relatively flat—at least for higher SNRs. However, ref. [37] also showed that at SNRs lower than 10 dB, training on varied oversampling rates can result in a significant decrease in accuracy (>10%) when compared to a network trained on a single oversampling rate. Figure 7 shows the impact of different training and testing sequence lengths at three different oversampling rates: 2, 4, and 8. Smaller oversampling rates resulted in better performance since more unique symbols would be seen for the same sequence length. An additional comparison was made in order to determine the impact of the oversampling rate when the number of observed symbols is held constant. Table 4 and Table 5 show the performance of the best networks trained on 128 and 1024 samples, respectively, for different oversampling rates and observed symbols. For the same number of observed symbols, a lower oversampling rate always resulted in a higher PCC. Initially, this seems to contradict the results found in Section 4 where samples consistently outperformed symbols. This suggests that the benefit of training samples rather than symbols is not the increased sequence length, but that the RNN is better able to examine the temporal relationship between consecutive samples than consecutive symbols. Comparing Table 4 and Table 5 shows that, for fewer observed symbols, training on a longer sequence length resulted in a small decrease in performance—up to 4%. However, when observing more symbols, training on a longer sequence resulted in a large increase in performance—up to 10%. This supports previous results suggesting that networks trained on smaller sequence lengths outperform networks trained on longer sequence lengths when tested on shorter sequence lengths. However, the opposite is also true and results in a significantly larger gap in performance. Based on these results, it is best to choose a training sequence length as close as possible to the one that will be used in testing. However, if the testing sequence length is not known, it is better to train on a longer sequence length.

### 5.5. Summary

Training on a smaller sequence length could save time in both data collection and network training. However, it is important to recognize that by training on a smaller sequence length, there is an increased risk of capping network performance due to insufficiently training the model to generalize to longer sequence lengths. The advantage of training on a longer sequence length increases with the complexity of the problem. For signals with a larger frequency offset, lower SNR, and variable oversampling rate, a network trained on a smaller sequence length for the same number of examples is unlikely to learn to generalize over this complicated problem space. As such, training a network with a larger sequence length will result in better generalization to unseen data. Future work will include training short sequence length networks with more examples to see if generalization can be improved.

The results of this section suggest that the sequence length used in training should be carefully considered. For less complex problems or for short signal bursts, training and evaluating on a short sequence length is likely preferable. However, for more complex problems or variable signal lengths, training on a longer sequence length will result in better generalization and allow for increased performance when evaluating longer input signals.

## 6. “Just Enough” Decision Making

The idea of “just enough” decision making, as introduced in our initial work [1], is that an RNN can be made more efficient by stopping its processing of new inputs once it is sufficiently confident in its decision. This is a separate concept from “early-stopping” which aims to prevent overfitting by stopping network training when certain criteria are met. In “just enough” decision making, no changes are currently made in the training stage—instead all analysis is carried out during inference. Prior work focused on reducing the number of samples processed, however, it did not allow for more samples to be processed than initially trained for. As such, this work will examine the impact of JED when allowed to extend past the number of samples trained for.

As the time-complexity of the network is linear with the input sequence length, the number of samples processed is used as a measure of the processing speed, while the true processing speed of the network will be dependent on hardware and the specific network architecture, it is worth noting that the post-processing required for JED requires significantly less time than a single time step of a 1-layer LSTM.

The JED approach relies on softmax values to make decisions. To show that softmax is a reasonable approximation of posterior probabilities, the softmax outputs of a network were compared with the ML classifier. The ML plots were generated by determining the percentage of times the ML classifier predicted each class for a given true class. The softmax plots were generated by finding the average softmax values for each class for a given true class. Both were averaged over 1000 examples per class for SNRs from 0 to 10 dB. Figure 8 shows the PCC for the maximum likelihood classifier and Figure 9 shows the average softmax value for the chosen network. When a signal is fed into the network, for each input a softmax value output for each of the five classes is given. The highest softmax value is chosen as the estimated class. The softmax outputs for each of the five classes were averaged over multiple examples to determine average trends when given different true signals. The outputs were taken from the **symbols** network with the highest PCC trained on 512 symbols) for each true class. The softmax values appear to present a reasonable approximation of the posterior probabilities—particularly for larger numbers of symbols, while this is true on average, at specific SNRs network bias can result in the softmax plots diverging from the expected trend. Values for 16QAM at 0 dB and 64QAM at 10 dB do not increase substantially as the number of observed symbols increases. Further examination of the network shows that it tends towards 16QAM at high SNRs and 64QAM at low SNRs, while softmax can be a useful approximate for posterior probabilities, it can be affected by network bias and should not be taken as a direct measure of confidence in the network. However, softmax values over time may be a reasonable measure of confidence. Future work should investigate JED as a measure of confidence as well as consider decision criteria that are not reliant on softmax values.

In our preliminary paper [1], four options were explored as decision criteria: threshold, subset, subset above threshold, and delta-threshold. As the delta-threshold method performed the best in nearly all cases, it is chosen for the following analysis. Figure 10 shows an overview of the delta-threshold technique for a single time-step. As JED is a post-processing technique, the actual model under consideration does not change. However, rather than getting outputs for all possible time-steps—which would still occur sequentially—the decision criteria is examined at each time step. If the decision criteria is met, then the final output is returned and no further inputs are processed. The delta-threshold (DEL) technique is designed to look for stability in the neural network’s output softmax value by examining its change over each input. The technique depends on two user-defined variables: *delta-threshold* which determines the amount of change between outputs that is tolerated and *duration*, the number of consecutive inputs for which the change must be below the *delta-threshold*. Two change values are used. The first is the change between the current input and the first input in the series. The second is the change between consecutive inputs. For example, if *duration* is 100, the *delta-threshold* is 0.1, and the first sample in the series has a softmax output of 0.9 for a given class, then the output of the network must stay between 0.8 and 1 for that class for 100 samples. Additionally, the change between consecutive samples must not exceed 0.1; if the output drops from 0.97 to 0.8, the counter will reset. The network will only make its decision if both these conditions are true for the highest class for 100 consecutive samples. However, to prevent the network from processing samples for an infinitely long time, a maximum number of samples to process was added. If the network has not made a decision by the time 2048 samples have been processed, it will output the current decision.

One limitation of the JED method is that it currently only works for classification and detection problems. For a time-series prediction problem like the one defined in [38] or an autoencoder used for classification like those seen in [5,39] alternative decision criteria would need to be developed.

### 6.1. Trade-Offs

Multiple JED combinations were tested for the different models. *Delta-thresholds* of 0.1, 0.2, 0.3, 0.4, and 0.5 and *durations* of 100, 200, 300, 400, 500, 600, 700, and 800 were tested. In each case the maximum number of samples that could be processed was set to 2048.

Figure 11 shows the results for the best model trained on 128 samples. As observed in [1], the value of *delta-threshold* had very little impact on accuracy, but did reduce the number of samples processed. For larger values of *delta-threshold*, fewer samples were processed. As seen in Section 5, performance seems to level out even when processing more samples on average. Accuracy peaked at 73% after processing an average of 1073 samples with a *duration* of 500 and a *delta-threshold* of 0.2. The accuracy when tested on a fixed length of 1024 was around 66%. For a similar number of average samples processed, the flexibility given by the JED method increase the PCC by almost 7%.

Figure 12 shows the results for the best model trained on 1024 samples. Again, the value of *delta-threshold* had very little impact on accuracy. Similarly, *delta-threshold* had little impact on the average number of samples processed, with the exception of when *delta-threshold* was set to 0.1, and the number was significantly higher. Performance increased slightly as *duration* increased and more samples were processed on average. Accuracy peaked at 75% after processing an average of 1522 samples with a *duration* of 800 and a *delta-threshold* of 0.1. The accuracy when tested on a fixed length of 1024 was around 67% while testing on a fixed length of 2048 was around 71%. The flexibility given by the JED method kept signals from being over-processed—due to exceeding their training range—which can result in an incorrect classification. By doing this, the PCC increased by 4% while processing less samples on average.

### 6.2. Nuisance

In addition to being shown to process fewer symbols overall, the JED method demonstrated its ability to differentiate between inputs based on signal complexity. For example, high SNR and BPSK signals were processed very quickly while lower SNR and higher order modulation signals were processed for a longer sequence length. A similar breakdown analysis to the one performed in [1] is performed below for the parameters addressed in Section 5—modulation, SNR, normalized frequency offset, and oversampling rate.

When determining which JED combination to use, we chose to optimize accuracy. As such, we selected the combination with the highest overall accuracy for the best 128 and 1024 models. However, for other applications, it may be preferable to accept some loss in accuracy (say 1%) to reduce the number of samples processed on average. To reflect that, and allow for a direct comparison between the best 128 and best 1024 models, each model was tested on two different JED combinations. Combination 1 resulted in the highest accuracy for the best model trained on 128 samples, and uses a *delta-threshold* of 0.2 and a *duration* of 500. Combination 2 resulted in the highest accuracy for the best model trained on 1024 samples, and uses a *delta-threshold* of 0.1 and a *duration* of 800. In all cases, the maximum number of samples that could be processed was 2048.

#### 6.2.1. Modulation

In Figure 13, for both the 128 and 1024 models, the accuracy and number of samples processed can be divided into three groups based on signal type. BPSK was processed for the least number of samples and had the highest accuracy in all cases, with the 1024 model having slightly higher accuracy than the 128 model while processing a similar number of average samples for each combination. QPSK and 8PSK were processed for a similar number of average samples and have similar accuracy for the 1024 model. However, for the 128 model 8PSK had significantly higher accuracy than QPSK even though it was processed for only slightly longer on average. This reflects the result shown in Figure 4 where the accuracy for QPSK did not improve much as more samples were processed. 16QAM and 64QAM were also processed for a similar number of samples for both the 128 and 1024 models. For the 128 model, accuracy was much higher for the 16QAM case than for the 64QAM case while the 1024 model the 64QAM case had higher accuracy than the 16QAM case. Both results are reflected in Figure 4. For both JED combinations, signals were successfully differentiated based on the signal type. Signal types with similar accuracy, and that are commonly misidentified as each other, were processed for a similar number of samples.

#### 6.2.2. SNR

Examining Figure 14 shows an interesting result. For the SNRs inside the training range, as SNR increased the number of samples processed decreased. For the above 10 dB SNRs which the networks were not trained for, the number of samples processed begins to flatline as processing more samples will have minimal effect. This is likely due to a combination of being outside the training range and that higher SNRs have diminishing returns on accuracy. However, the same does not happen for the sub-zero dB range. As established previously, the 1024 model did a better job of generalizing to unseen data. This can be seen from the plot where more samples were processed for −5 dB than for 0 dB. In contrast, the 128 model actually processed less data for the −5 dB case than for the 0 dB case. As this occurred for both JED combinations, the difference is likely because the 128 model was “confidently wrong” at sub-zero SNR which allowed it to fulfill the JED requirements while having very poor accuracy, while performance is slightly higher at −5 dB for the 1024 model than for the 128 model, it came at the cost of processing the signal for a significantly longer period. Despite being processed for so much longer, performance is still barely above random guessing. Although JED can potentially improve accuracy and reduce the number of samples processed in some cases, it comes at the cost of relinquishing direct control over the number of samples processed.

#### 6.2.3. Frequency Offset

Examining Figure 15 shows that the performance curves for both the 128 and 1024 models are very similar to the 2048 curve shown for those models in Figure 6. However, significantly fewer samples were processed on average. Again, there is very little difference between the accuracy for the two JED combinations tested. However, there is a large gap in average samples processed due to the *duration* chosen. Unlike with Figure 14 there is not a large difference in the number of samples processed between the 128 and 1024 models for data outside the training range.

#### 6.2.4. Oversampling Rate

Examining Figure 16 shows a slight difference between the two JED combinations. Combination 2 performs slightly better than combination 1, particularly for higher oversampling rates. However, combination 2 processed significantly more samples on average. Surprisingly, for the 128 case with combination 1, the network actually processed slightly more samples on average for an oversampling rate of 2 than for an oversampling rate of 4 for both combinations. However, an oversampling rate of 2 still had higher accuracy. For the 1024 case, the network processed more samples for an oversampling rate of 4 than an oversampling rate of 2, as expected. More samples were processed for an oversampling rate of 8 in both cases.

#### 6.2.5. Summary

The different JED combinations had very little effect on accuracy; overall the PCC curves looked very similar to the 2048 curves shown in Section 5. However, the combination had a significant impact on the number of samples processed, while the shape of each samples processed curve for combination 2 is very similar to that of combination 1, the actual number of samples processed is consistently higher due to the higher value of *duration* chosen.

Using the JED method can result in both higher accuracy and fewer samples processed on average and can be very useful when the evaluation sequence length is unknown or when there is a lot of variation in the input data. However, using JED relinquishes direct control of the number of samples processed. In some, cases this may result in processing many more samples with little gain. When processing sub-zero SNR data for a sufficiently trained network, the JED method processed a large number of samples, but achieved very little improvement in performance. This problem can be minimized by choosing an appropriate value of *delta-threshold*, *duration*, and maximum sequence length.

## 7. Discussion

This paper aimed to “decouple” training and testing sequence lengths for spectrum sensing networks. Section 4 demonstrated that the accuracy of networks trained on short sequence lengths can continue to improve when tested on longer sequence lengths. To better illustrate this, a comparison was done with a maximum-likelihood classifier. The maximum-likelihood classifier under ideal conditions forms the upper bound on classification accuracy. However, due its computational complexity and its assumptions about synchronization and matched filtering, it is not used in many practical applications. When testing on longer sequence lengths, the networks were able to achieve accuracy higher than the ML classifier did with the original training length. This proves that a trained AMCs PCC is bound by the sequence length used in testing rather than the one used in training.

While the number of observed symbols is what ultimately bounds the network performance, the number of symbols used in training is still relevant. Notably, the network trained on 512 symbols was closer to the ML bound than the network trained on 64 symbols—even though both observed 512 symbols in training. A training sequence length much shorter than the one used in testing may result in decreased performance.

While Section 4 proved that “decoupling” is achievable, it only addressed an ideal case. To further determine the impact of training and testing sequence lengths a more complex case with frequency offset and varied oversampling rates was considered in Section 5. Initial results showed diminishing returns when testing some networks on longer sequence lengths. The performance of networks trained on short sequence lengths had a tendency to saturate when tested on significantly longer sequence lengths. To gain a better understanding of this behaviour the impact of several different parameters was examined. Of particular interest was an examination of how training sequence length impacted network bias and generalization to data outside the training range.

As networks trained on shorter sequence lengths observe fewer symbols in training, there were concerns that they would be more likely to develop a bias. However, there was no clear trend found between training on a shorter sequence length and a higher likelihood of network bias. Despite this bias is still impacted by sequence length. When testing a biased network on a longer sequence length—regardless of the sequence length used in training—the bias actually becomes more apparent.

Examining network performance across each modulation scheme also provided some insight into the saturation observed. For the networks trained on a short sequence length, performance did not increase evenly across all signal types. Instead as the testing sequence length increased, the network began to prefer certain classes over others. In contrast, the network trained on a long sequence length saw performance improvement for all classes, while this may seem to be simply the result of a biased network, the network was not biased as it was not choosing one class to the detriment of another. In the biased networks that were examined, performance for some classes actually saw a significant decrease—in this case it simply stays the same. Ultimately, this suggests that the networks trained on a smaller sequence length did not adequately learn to identify each modulation. As a result, they were unable to generalize to longer sequence lengths.

Testing the networks on data outside the training range showed similar results. Networks trained on shorter sequence lengths were unable to generalize to unseen data. In contrast, networks trained on longer sequence lengths saw significant improvement in performance when sequence length increased—even for data outside the training range.

The goal of “decoupling” was to allow the testing and training sequence length to be set independently rather than assuming them to be equivalent. However, they are not truly independent. To achieve the best performance the training sequence length should be chosen so that it is as close as possible to the sequence lengths that will be seen in testing. In cases where the testing sequence length is unknown or variable, the best choice of training sequence length will depend on the complexity of the problem. A smaller training sequence length will save time in both data collection and network training. However, it comes with an increased risk of capping network performance due to insufficiently training the model to generalize to longer sequence lengths. The advantage of training on a longer sequence length increases with the complexity of the problem.

While Section 5 showed how to choose an appropriate training sequence length, Section 6 addressed how to handle a variable sequence length in training. In applications that handle signals of different lengths or have a signal of interest of unknown length, it is unclear when the network should stop processing new inputs and output a decision. One potential approach to this is the JED method introduced in our preliminary work [1]. The JED method dynamically chooses how much sequential input data to process based on signal complexity. In addition to allowing the network to deal with signals of unknown or variable sequence lengths, it can also benefit time-sensitive applications. Electronic warfare, radar, and dynamic spectrum access systems need to make decisions as quickly and accurately as possible. The JED approach allows a decision to be returned before the entirety of the signal is processed without sacrificing accuracy.

The JED method is solely accomplished in post-processing and is based on user-defined values. The values chosen can be “tuned” based on the application. The combinations used were chosen so as to maximize accuracy, however, in other applications it may be preferable to instead further reduce the number of samples processed. As it stands, the first combination improved accuracy by almost 7% while processing a similar number of samples and the second combination improved accuracy by 4% while processing around 500 fewer samples on average.

To show the utility of the JED method, multiple parameters were examined to see how they impacted both the number of samples processed and the accuracy of the network. In general, the results showed that the JED method processed simpler signals for shorter sequence lengths. However, when the method saw data that had a lower SNR than seen in training, different networks gave different responses. The first network was confidently incorrect, so the average number of samples processed decreased even though the problem was more difficult. The second network had better generalization, so the average number of samples processed continued to increased as SNR decreased, while the second network was closer to the stated goal of JED, its accuracy at low SNRs was not much better than the first network. Twice as many samples were processed on average, but performance was still barely above random guessing. Relinquishing direct control of the sequence length can result in processing many more samples with no real benefit. This problem can be minimized by choosing an appropriate value for the user-defined parameters—particularly the maximum sequence length.

The JED method provides a useful decision criteria for applications where the signal length in inference is unknown or variable. It can also be beneficial for time-sensitive applications like electronic warfare and dynamic spectrum access. However, it is most useful when there is significant variation in the input data. If some signals are significantly easier to identify than others, using JED post-processing can result in both higher accuracy and fewer samples processed on average. However, using JED relinquishes direct control of the number of samples processed which may not be desirable in some applications.

## 8. Conclusions and Future Work

In applications like dynamic spectrum access, radar, or electronic warfare the sequence length of a signal of interest may be variable or completely unknown. Networks generated for these applications may need to be adapted to sequence lengths different than they were trained for. As such, it is important to consider the impact of training and testing sequence lengths individually. “Decoupling” training and testing sequence lengths allows us to better determine a training length in the absence of conclusive knowledge regarding the sequence length that will be used in inference.

While most spectrum sensing approaches have used a fixed sequence length in inference, this approach has many downsides. If the length of the signal of interest is variable or unknown, it can be difficult to choose a reasonable testing sequence length. Even if the testing sequence length is known, a fixed value will result in all input signals being processed for the same number of samples—regardless of complexity. Instead, it would be preferable to only process signals for as long as is necessary to confidently identify them. This capability is particularly valuable in time-sensitive applications like dynamic spectrum access and electronic warfare.

To dynamically alter the input sequence length, the “just enough” decision making delta-threshold technique was used, which examines the change in the softmax value of the output over time allowing for fewer observations to be processed. By using this technique, accuracy was actually improved when compared to training and testing on a fixed sequence length as the network was able to dynamically alter the number of observed samples based on signal complexity. Although using the JED method improved performance and reduced evaluation time, it also relinquished direct control of the testing sequence length which can result in processing many more samples for very little gain.

As the metric used in “just enough” decision making was able to differentiate between complex and simple signals, it could prove useful as a confidence metric. Future work will include analysis of its potential as a confidence metric as well as training on varying sequence lengths to try and achieve better generalization.

## Figures and Tables

**Figure 1 sensors-22-04706-f001:**
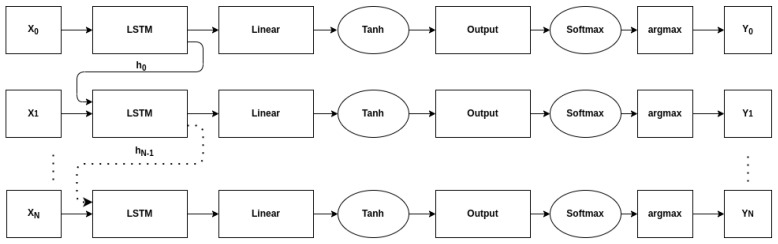
Diagram showing the general architecture of the considered models.

**Figure 2 sensors-22-04706-f002:**
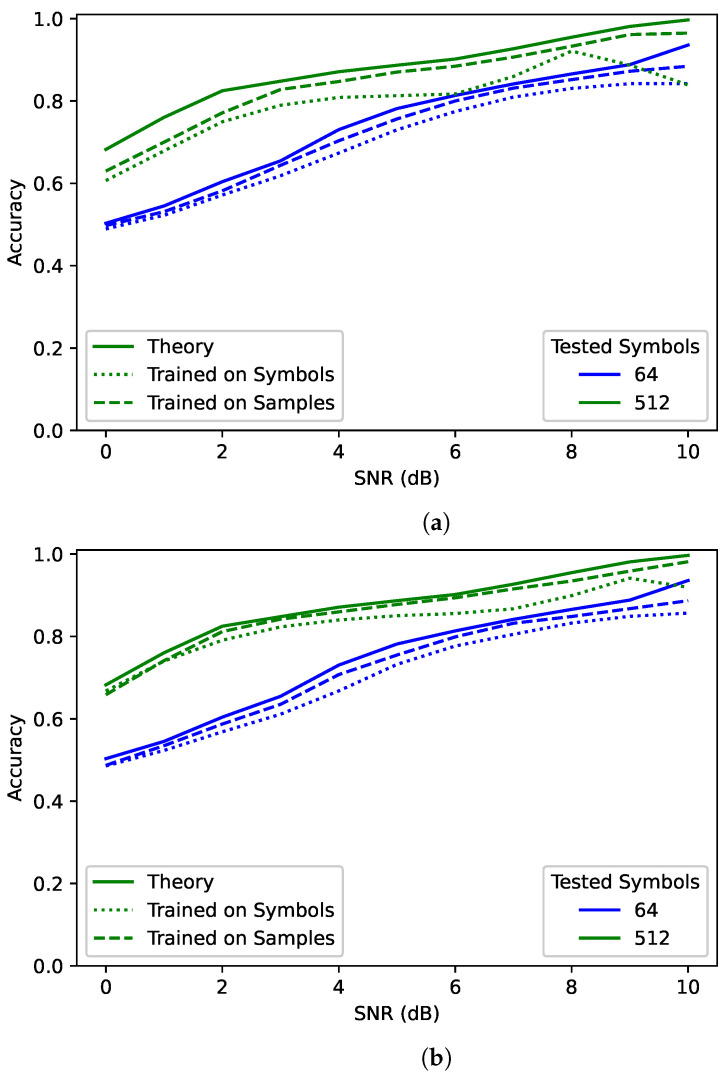
A comparison between the performance upper bound and the best **samples** and **symbols** networks trained on (**a**) 64 symbols and (**b**) 512 symbols.

**Figure 3 sensors-22-04706-f003:**
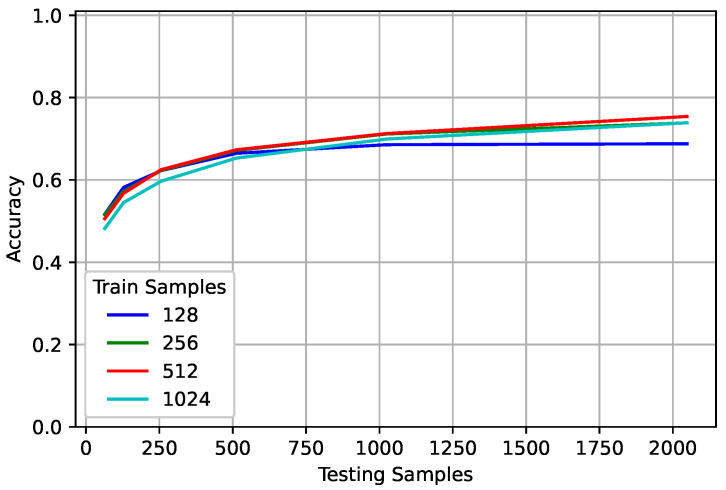
The **nuisance** networks trained on different sequence lengths were compared to each other for different testing sequence lengths. For the same trained sequence length, evaluating on a longer sequence length increased the PCC with diminishing returns. For the same sequence length in evaluation, networks trained on a sequence length close to the the evaluation sequence length typically performed better.

**Figure 4 sensors-22-04706-f004:**
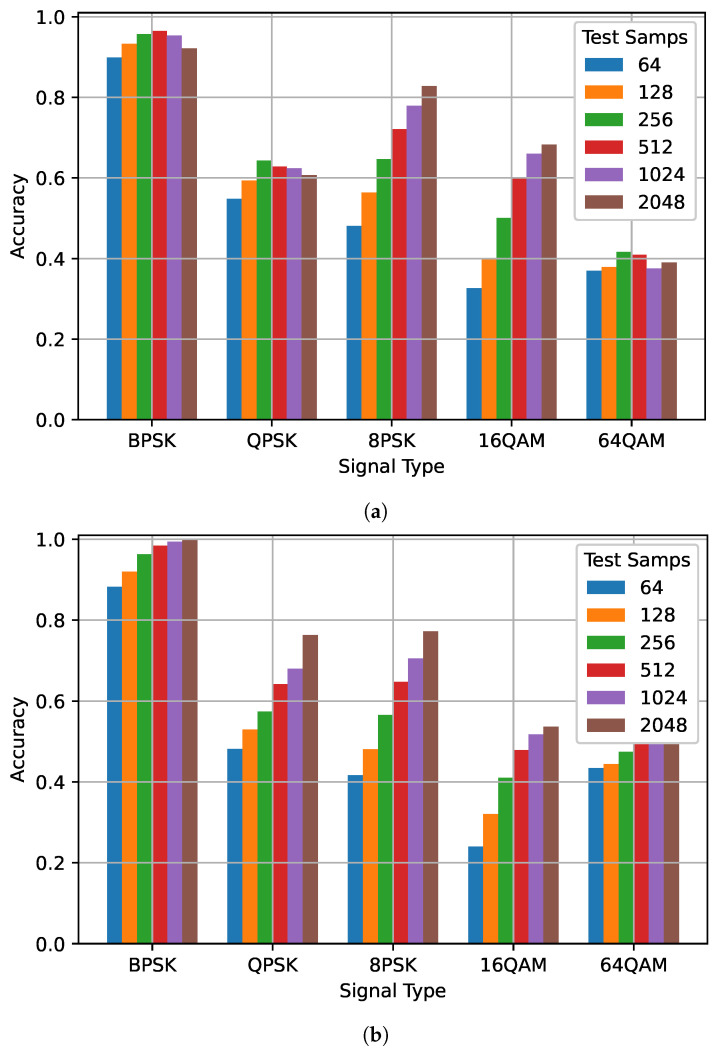
The impact of different training and testing sequence lengths on performance for different modulations on the best network trained on (**a**) 128 samples and (**b**) The impact of 1024 samples. Sequence lengths of 64, 128, 256, 512, 1024, and 2048 were tested for both networks.

**Figure 5 sensors-22-04706-f005:**
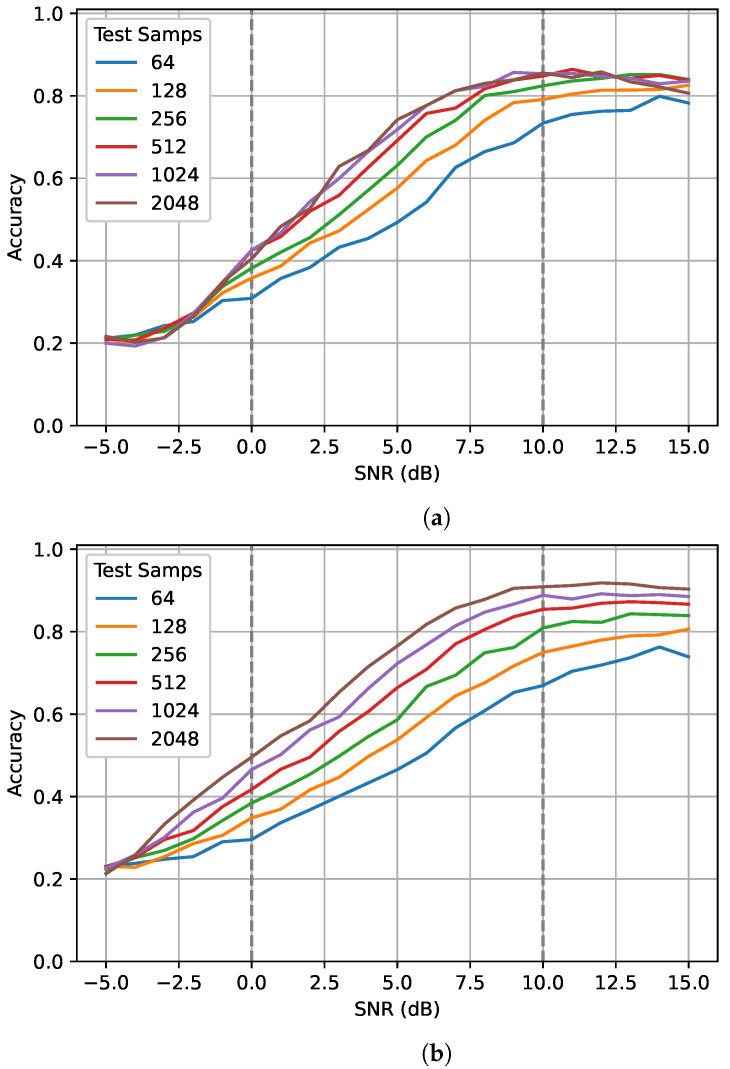
The impact of different training and testing sequence lengths on performance for different SNRs. Sequence lengths of 64, 128, 256, 512, 1024, and 2048 were tested on the best network trained on (**a**) 128 samples and (**b**) 1024 samples. The gray dotted lines show the SNR training range.

**Figure 6 sensors-22-04706-f006:**
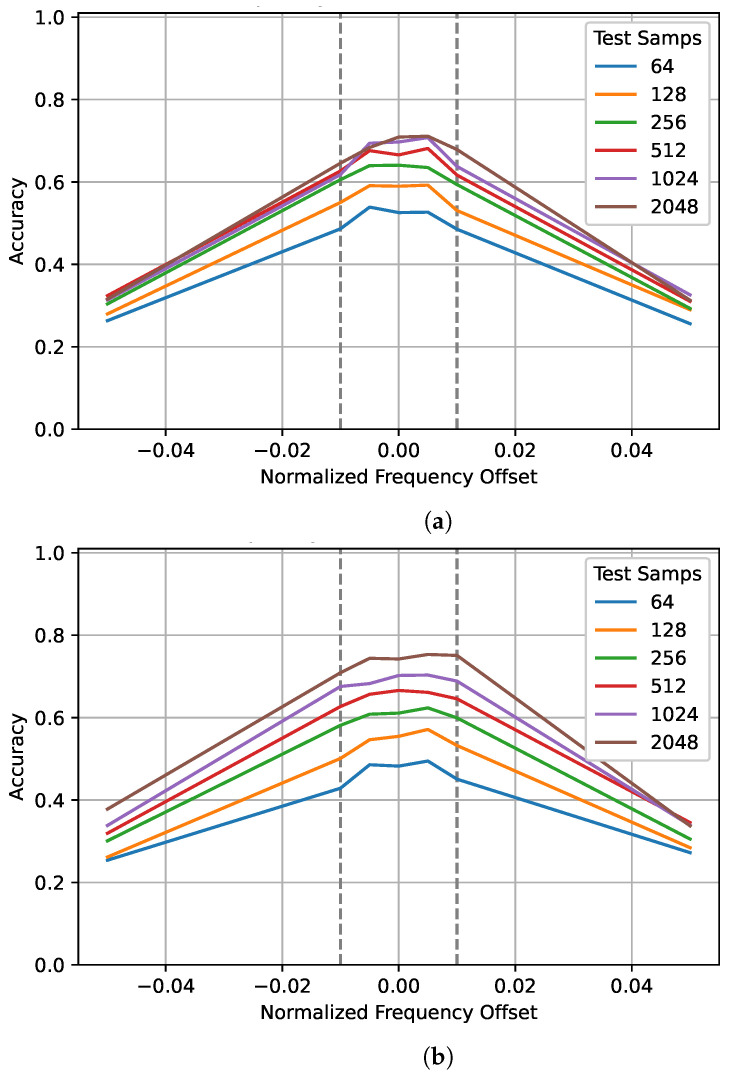
The impact of different training and testing sequence lengths on performance for different normalized frequency offsets. Sequence lengths of 64, 128, 256, 512, 1024, and 2048 were tested on the best network trained on (**a**) 128 samples and (**b**) 1024 samples. The gray dotted lines show the frequency training range.

**Figure 7 sensors-22-04706-f007:**
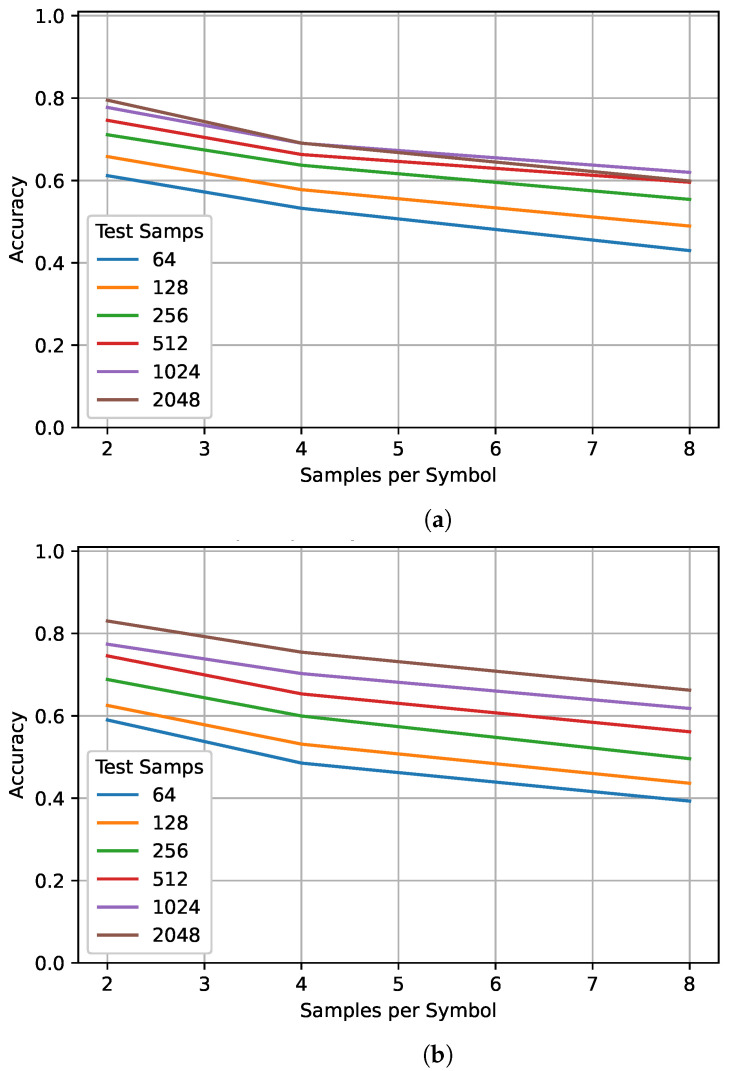
The impact of different training and testing sequence lengths on performance for different oversampling rates. Sequence lengths of 64, 128, 256, 512, 1024, and 2048 were tested on the best network trained on (**a**) 128 samples and (**b**) 1024 samples.

**Figure 8 sensors-22-04706-f008:**
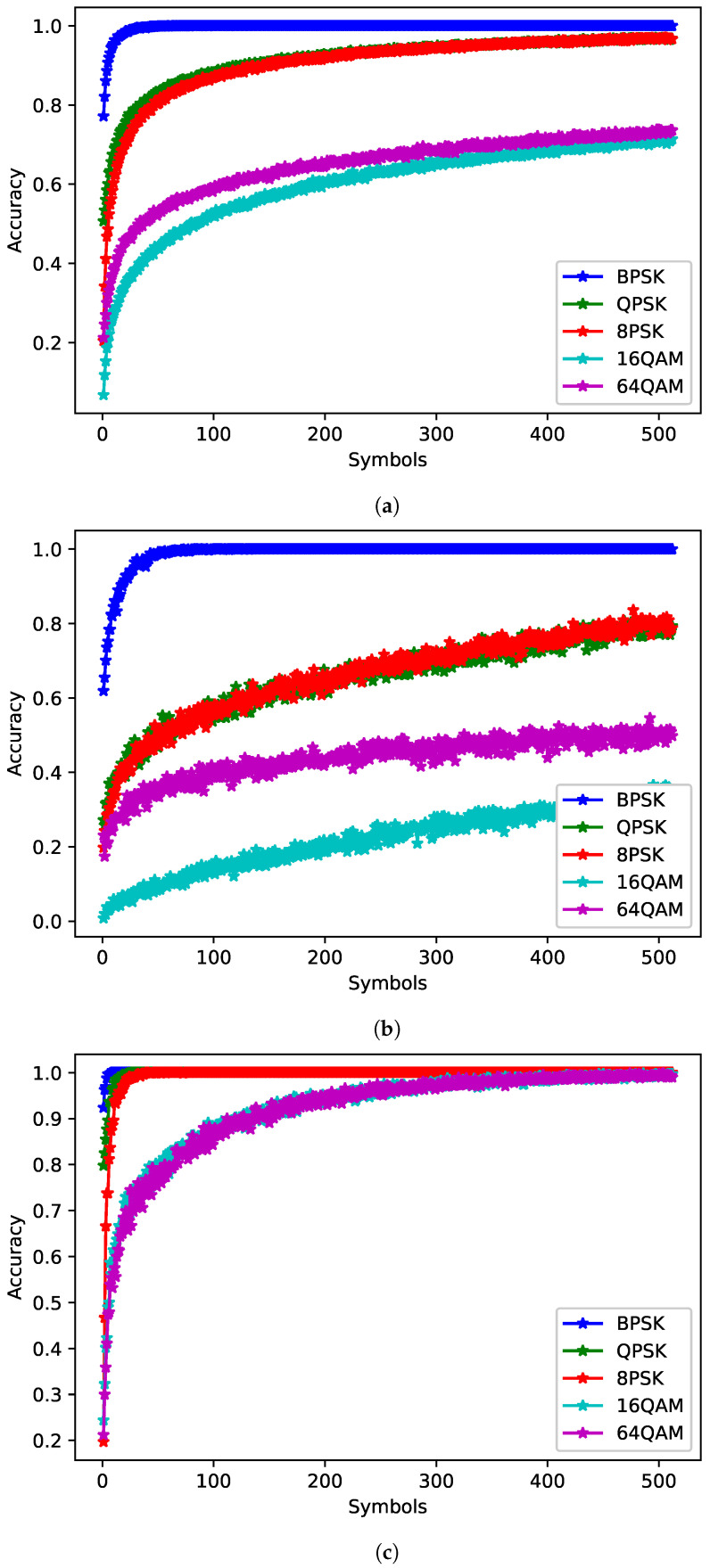
The ML estimate of the posterior probabilities (**a**) averaged from 0 to 10 dB (**b**) at 0 dB and (**c**) at 10 dB.

**Figure 9 sensors-22-04706-f009:**
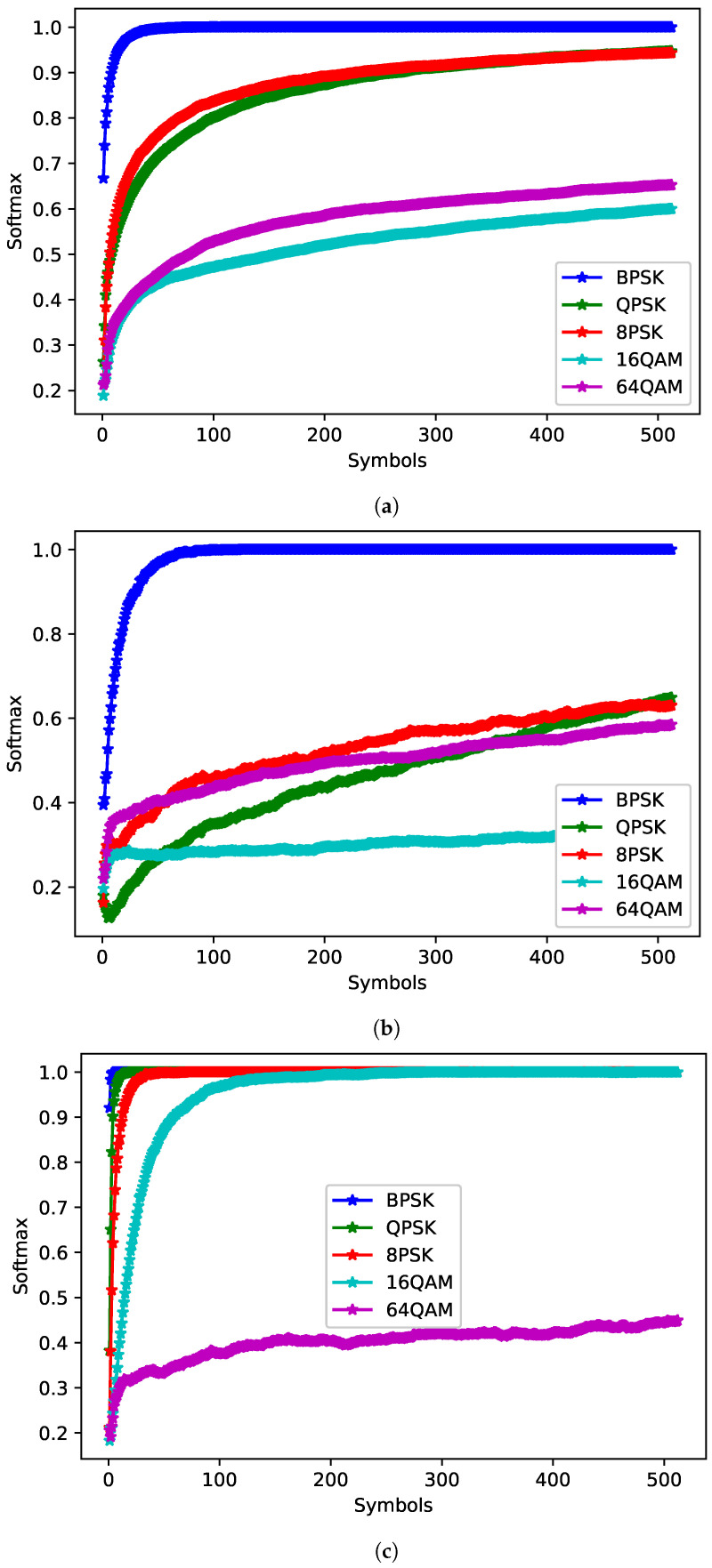
The average softmax outputs of the best **symbols** network trained on 512 symbols (**a**) averaged from 0 to 10 dB (**b**) at 0 dB and (**c**) at 10 dB.

**Figure 10 sensors-22-04706-f010:**
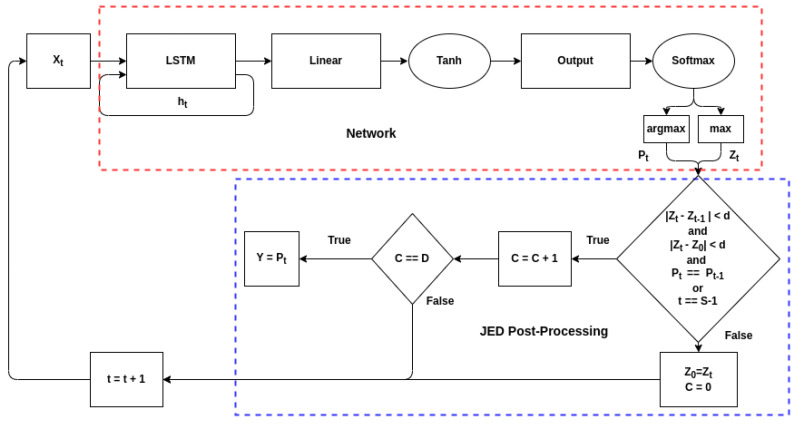
A diagram of the JED process where d is *delta-threshold*, D is *duration*, and S is the maximum number of samples that can be processed.

**Figure 11 sensors-22-04706-f011:**
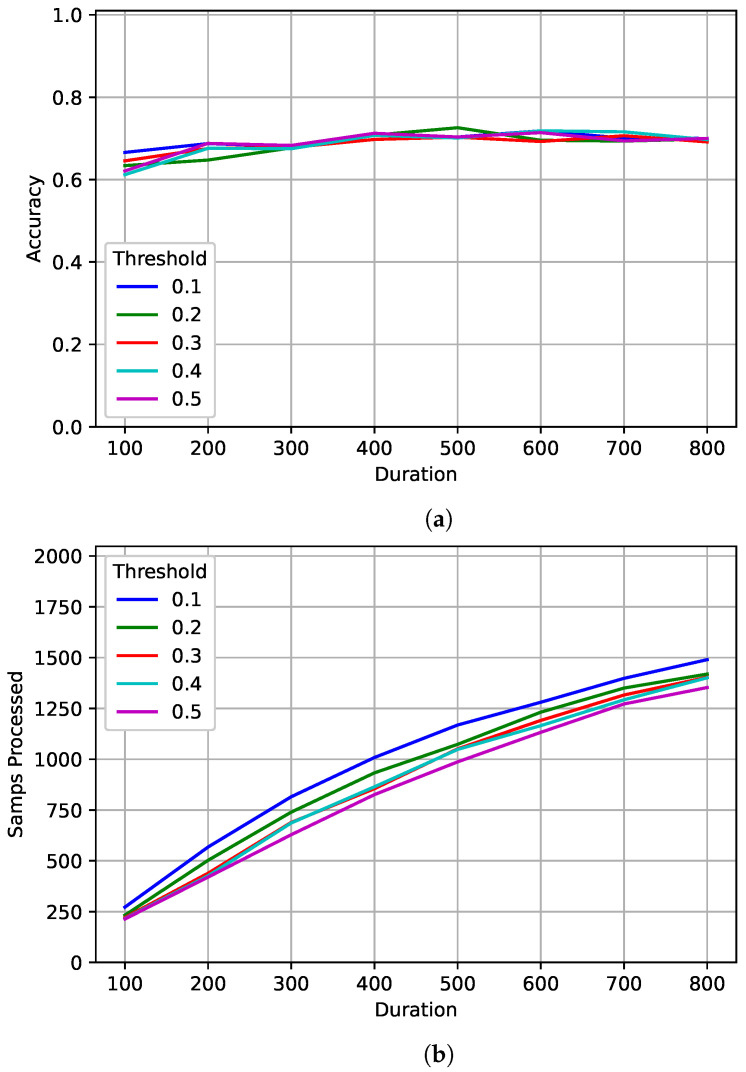
A tradeoff analysis of (**a**) accuracy and (**b**) the number of samples processed of the different JED parameters for the best models trained on 128 samples.

**Figure 12 sensors-22-04706-f012:**
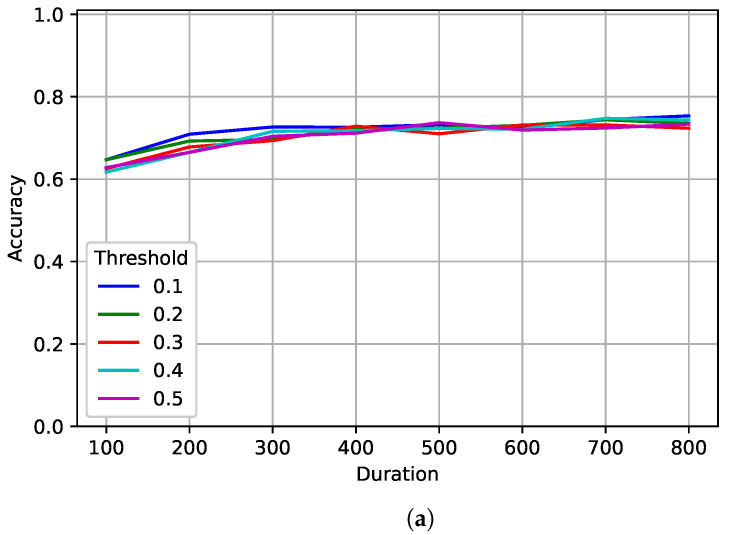
A tradeoff analysis of (**a**) accuracy and (**b**) the number of samples processed of the different JED parameters for the best models trained on 1024 samples.

**Figure 13 sensors-22-04706-f013:**
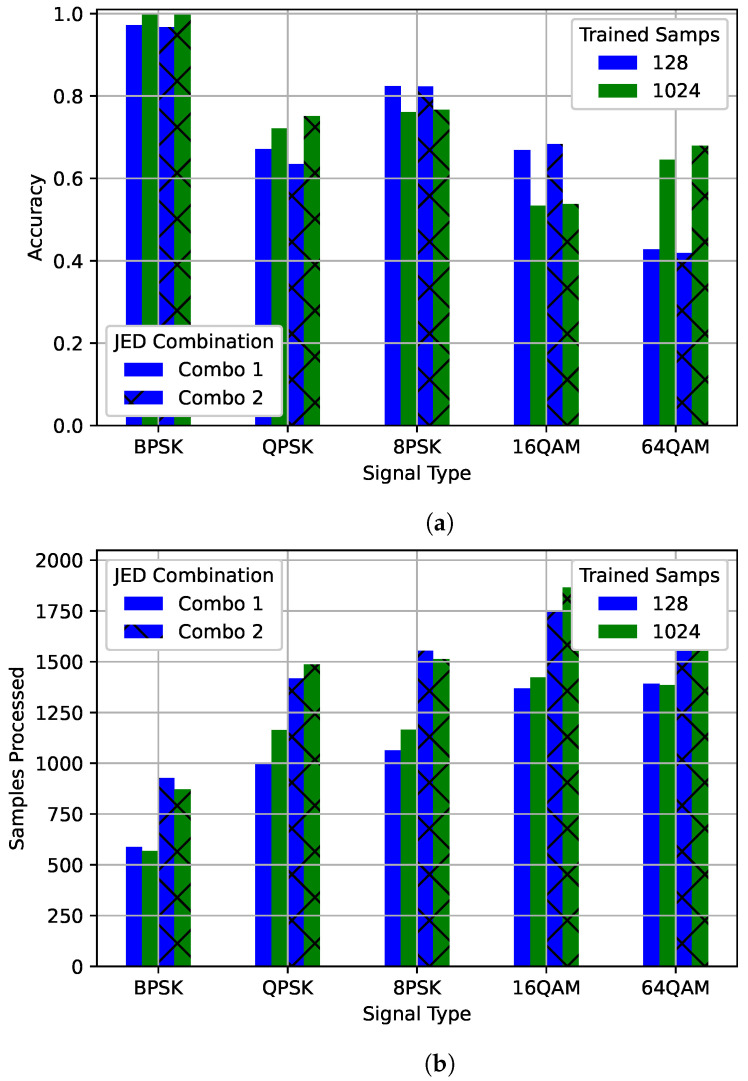
An examination of (**a**) accuracy and (**b**) the number of samples processed of JED for different modulations. The best model for 128 and 1024 are each shown after processing with two different JED combinations.

**Figure 14 sensors-22-04706-f014:**
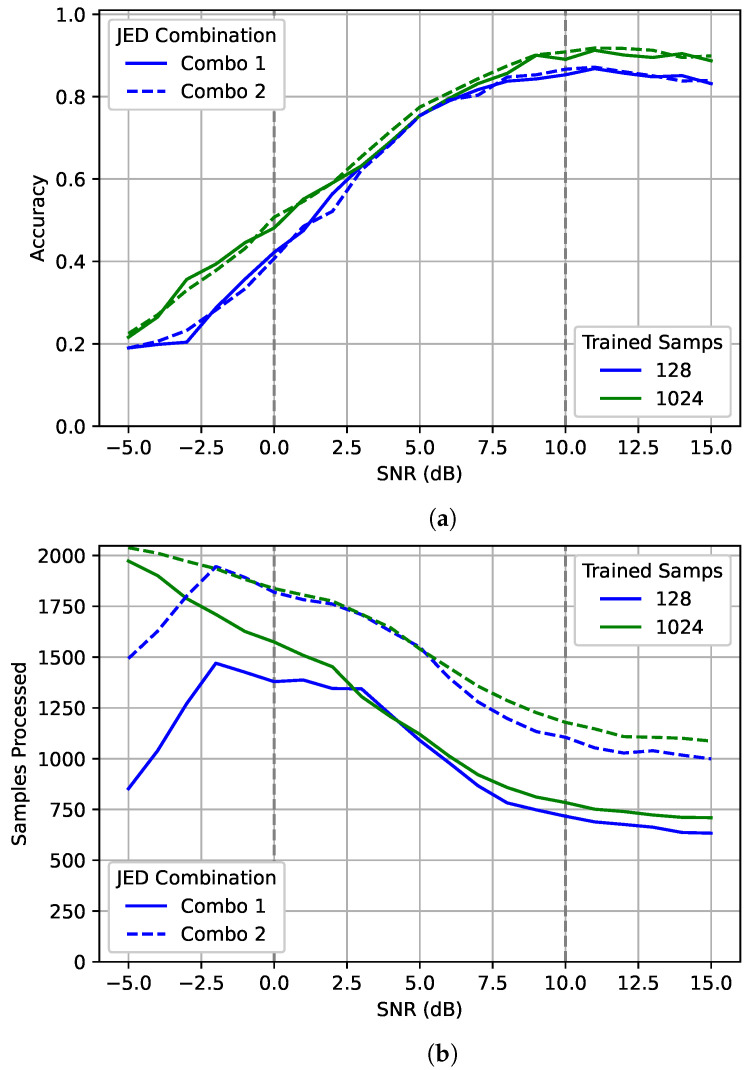
An examination of (**a**) accuracy and (**b**) the number of samples processed of JED for different SNRs. The best model for 128 and 1024 are each shown after processing with two different JED combinations. The gray dotted lines show the training range for SNR.

**Figure 15 sensors-22-04706-f015:**
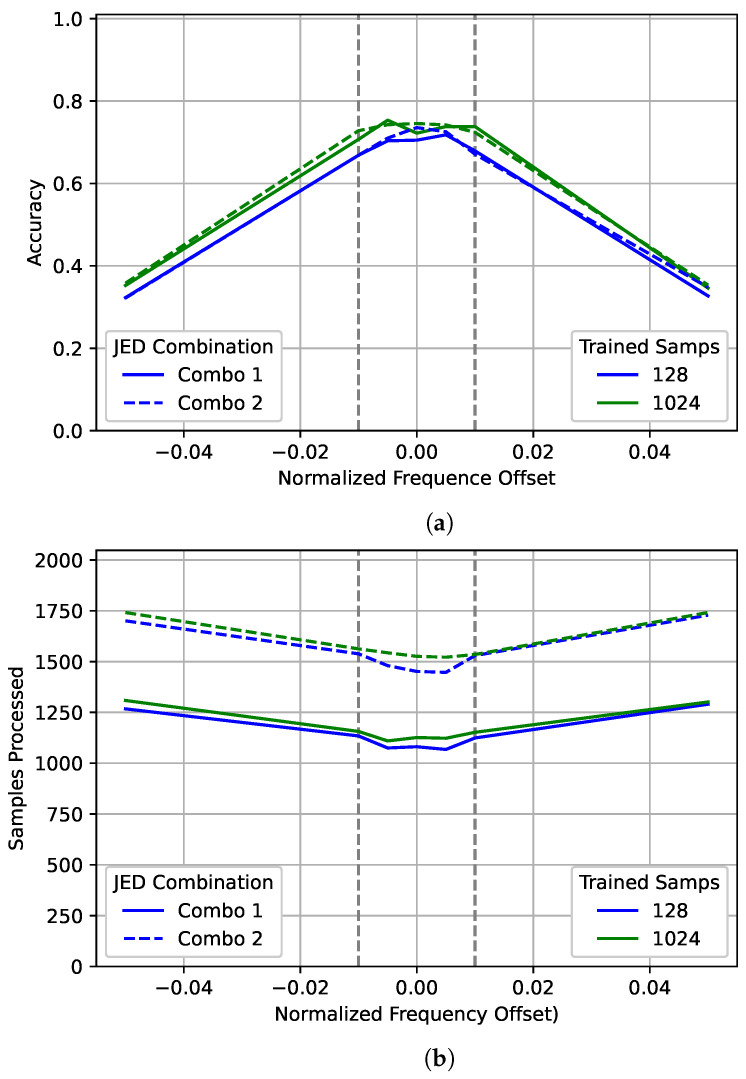
An examination of the (**a**) accuracy and (**b**) the number of samples processed of JED for different normalized frequency offsets. The best model for 128 and 1024 are each shown after processing with two different JED combinations. The gray dotted lines show the training range the frequency offset.

**Figure 16 sensors-22-04706-f016:**
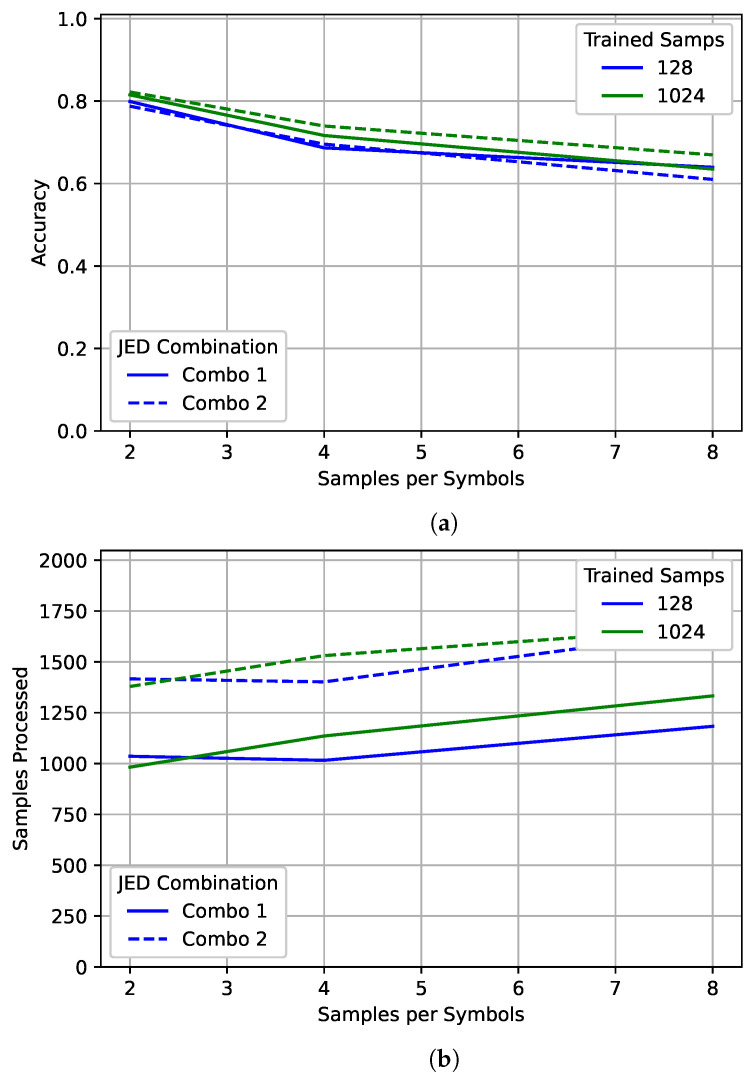
An examination of (**a**) accuracy and (**b**) the number of samples processed of JED for different oversampling rates. The best model for 128 and 1024 are each shown after processing with two different JED combinations.

**Table 1 sensors-22-04706-t001:** RNN input assumptions for each considered spectrum scenario.

	Symbol	Sample	Nuisance
**Input Type**	symbols	samples	samples
**Oversampling Rate**	2	2	2, 4, 8
**Frequency Offset**	0	0	±0.01
**Number of Examples**	1,000,000	1,000,000	2,000,000

**Table 2 sensors-22-04706-t002:** General architecture of the considered models.

Name	Layers	Input	Output	Weights
LSTM + dropout ^1^	l∈(1,4)	2	h∈(15,512)	4h(2hl−h+l+2)
Dense + Tanh	1	*h*	d∈(15,512)	hd
Dense + Softmax	1	*d*	5	5d

^1^ Dropout probability of 0.5.

**Table 3 sensors-22-04706-t003:** $F_{1}$ score on different sequences for biased networks.

Train/Test	BPSK	QPSK	8PSK	16QAM	64QAM
**128/128**	0.914	0.509	0.542	0.326	0.454
**128/1024**	0.967	0.676	0.671	0.300	0.601
**1024/128**	0.899	0.415	0.334	0.345	0.394
**1024/1024**	0.995	0.602	0.146	0.500	0.575

**Table 4 sensors-22-04706-t004:** PCC of observed symbols for networks trained on 128 samples.

	Oversampling Rate		
Observed Symbols	2	4	8
**8**	-	-	0.421
**16**	-	0.508	0.487
**32**	0.604	0.572	0.532
**64**	0.651	0.628	0.574
**128**	0.695	0.662	0.581
**256**	0.729	0.680	0.576
**512**	0.725	0.708	-
**1024**	0.731	-	-

**Table 5 sensors-22-04706-t005:** PCC of observed symbols for networks trained on 1024 samples.

	Oversampling Rate		
Observed Symbols	2	4	8
**8**	-	-	0.393
**16**	-	0.485	0.436
**32**	0.590	0.531	0.496
**64**	0.625	0.600	0.561
**128**	0.688	0.653	0.618
**256**	0.745	0.703	0.662
**512**	0.774	0.755	-
**1024**	0.830	-	-

## Data Availability

Not applicable.

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
