# Peer review of "Decoupling RNN Training and Testing Observation Intervals for Spectrum Sensing Applications"

_sensors, 2022, doi:10.3390/s22134706_

Round 1
Reviewer 1 Report
This paper illustrates the benefits and considerations needed when “decoupling" these observation intervals for spectrum sensing applications, using modulation classification as the example use case. In particular, this work shows that, intuitively, recurrent neural networks can be leveraged to process less data (i.e. shorter observation intervals) for simpler inputs (less complicated signal types or channel conditions). Less intuitively, this works shows that the ‘’decoupling" is dependent on appropriate training to avoidbias and ensure generalization.
In this article, the recurrent neural networks is mentioned in many places.However, there is no mathematical discussion and analysis on the recurrent neural networks, and even there is no any mathematical discussion and analysis on the discussed problems and improved problems. But there is only some numerical simulations and numerical experiments are used to get some results and the obtained results are not strictly proved. Moreover, there still exist some problems to be addressed:
1. What is the meaning of ``decoupling"? Explain it in the Introduction Section or give a definition.
2. The main contribution and novelty of this paper shall be summarized in Introduction.
3. Some remarks should be added to show the advantages of this paper comparing with the existing references.
4. The usage of punctuation in the whole mathematical formula of this paper are not standard.
Reviewer 2 Report
The manuscript entitled “Decoupling RNN Training and Testing Observation Intervals for Spectrum Sensing Applications” has been investigated in detail. The topic addressed in the manuscript is potentially interesting and the manuscript contains some practical meanings, however, there are some issues which should be addressed by the authors:
1) In the first place, I would encourage the authors to extend the abstract more with the key results. As it is, the abstract is a little thin and does not quite convey the interesting results that follow in the main paper. The "Abstract" section can be made much more impressive by highlighting your contributions. The contribution of the study should be explained simply and clearly.
2) The readability and presentation of the study should be further improved. The paper suffers from language problems.
3) The “Introduction” section needs a major revision in terms of providing more accurate and informative literature review and the pros and cons of the available approaches and how the proposed method is different comparatively. Also, the motivation and contribution should be stated more clearly.
4) The importance of the design carried out in this manuscript can be explained better than other important studies published in this field. I recommend the authors to review other recently developed works.
5) “Discussions” section should be added in a more highlighting, argumentative way. The author should analysis the reason why the tested results is achieved.
6) The authors should clearly emphasize the contribution of the study. Please note that the up-to-date of references will contribute to the up-to-date of your manuscript. The study named- Crude oil time series prediction model based on LSTM network with chaotic Henry gas solubility optimization- can be used to explain the method in the study or to indicate the contribution in the “Introduction” section.
7) The complexity of the proposed model and the model parameter uncertainty are not enough mentioned.
8) The effect of the parametric uncertainty is not discussed in detail. How did the comparison methods perform with or without the uncertainty?
9) It will be helpful to the readers if some discussions about insight of the main results are added as Remarks.
This study may be proposed for publication if it is addressed in the specified problems.
Reviewer 3 Report
This work considers the RNN for the modulation classification application. There are some questions as follows:
1) Please provide the explanation of the “decoupling" problem that the paper considered in the Abstract? The motivations are not clear.
2) For the system mode, it is better to plot some figures in which the RNN can be applied.
3) For modulation classification, the CNN is shown to have better accuracy than RNN, moreover, the ML provides higher accuracy than RNN in Fig. 1, what is the significance of the considered RNN?
4) What are the differences between the conventional RNN and the proposed RNN architecture in Table 2?
5) For decision-making of time-sensitive applications such as spectrum sensing, how much faster is an RNN-based spectrum sensing application than a conventional one?
Round 2
Reviewer 1 Report
This article is now acceptable.
Reviewer 2 Report
All my comments have been thoroughly addressed. It is acceptable in the present form.
Reviewer 3 Report
The authors have addressed the raised concerns satisfactorily. I have no further comments.